# Person-centred eHealth intervention for patients on sick leave due to common mental disorders: study protocol of a randomised controlled trial and process evaluation (PROMISE)

Matilda Cederberg [1,2] Lilas Ali [1,2,3] Inger Ekman,[1,2,4] Kristina Glise,[5] Ingibjörg H Jonsdottir,[5,6] Hanna Gyllensten [1,2] Karl Swedberg,[1,2,7] Andreas Fors [1,2,8]

**Correspondence to**
Matilda Cederberg;
matilda.cederberg@gu.se

## ABSTRACT

**Introduction** The number of people dealing with common mental disorders (CMDs) is a major concern in many countries, including Sweden. Sickness absence resulting from CMDs is often long-lasting and advancing return to work is a complex process impacted by several factors, among which self-efficacy appears to be an important personal resource. Person-centred care (PCC) has previously shown positive effects on self-efficacy however this needs to be further investigated in relation to patients with CMDs and in an eHealth context.

**Methods and analysis** This study is an open randomised controlled trial comparing a control group receiving standard care with an intervention group receiving standard care plus PCC by telephone and a digital platform. The primary outcome measure is a composite score of changes in sick leave and self-efficacy. Participants will include 220 primary care patients on sick leave due to CMDs and data will mainly be collected through questionnaires at baseline and 3, 6, 12 and 24 months from the inclusion date. Inclusion is ongoing and expected to be completed during the fall of 2020. A process and health economic evaluation will also be conducted.

**Ethics and dissemination** This study was approved by the Regional Ethical Review Board in Gothenburg, Sweden. Results will be published in peer-reviewed scientific journals and presented at national and international scientific conferences. This project is part of a broader research programme conducted at the Gothenburg Centre for Person-Centred Care (GPCC), where extensive work is undertaken to disseminate knowledge on and implementation of PCC.

**Trial registration number** NCT03404583.

## Strengths and limitations of this study

► The intervention contains multiple components and is based on person-centred ethics.
► The randomised controlled trial will be evaluated for effectiveness, health economic outcomes and process evaluation.
► Data will be analysed using quantitative and qualitative methods.
► The intervention has a generic design which makes it applicable also to other conditions, and to various categories of healthcare professionals.
► Because the study is mainly conducted in a research centre, future adaptions may be needed before implementing the intervention in other contexts and settings.

estimated at €7.7 billion annually, of which about 50% is attributable to direct medical costs and 50% to work absence and out-of-work benefits.[1] The number of sick leaves with origins from mental illness has increased during the past decade, and it is now the most common cause of sick leave in Sweden.[4 5] The conditions most frequently causing sick leave are depression, anxiety and stress-related disorders, which are generally referred to as common mental disorders (CMDs).[2 6 7] Sick leave attributable to CMDs has a long mean duration, with high risk of recurrent sickness episodes.[7 8] In Sweden, the majority of consultations concerning CMDs take place in primary care, and most patients are treated within primary care.[1 9] First-line treatment often consists of cognitive behavioural therapy, medication or both,[10] but the relationship between sickness duration and return to work (RTW) is complex.[11–14] The existing evidence on interventions affecting

## INTRODUCTION

As a major determinant of short-term and long-term sick leave in many countries, mental illness bears a substantial portion of healthcare expenses and challenges.[1–3] In Sweden, the total cost of mental illness has been

RTW is limited; however, interventions that include a workplace component are more successful in improving RTW.[15–19] In addition, interventions with multiple components appear more successful than single-component interventions.[19] A recent systematic review by Etuknwa *et al*[20] evaluated which personal and social factors impacted RTW and found that self-efficacy, a positive attitude and support from leaders and co-workers were most relevant. The possible influence of self-efficacy on RTW has also been supported elsewhere.[21 22] Self-efficacy as a concept is rooted in social cognitive theory, which draws conclusions on human agency and its internal and external determinants.[23] Perceived self-efficacy is a personal judgement of an individual's capacity to handle general or specific situations in life, and as such, it is a core function of human agency. It is shaped by different sources of information and contextual factors; however, the best way to influence a person's self-efficacy is through successful experiences of mastery.[24] Moreover, a strong sense of self-efficacy can reduce vulnerability to stressors and increase resilience to cope with adverse events.[23 25]

Self-efficacy is a central notion in person-centred care (PCC).[26–29] PCC can be conceptualised as initiatives advocating that healthcare systems need to treat and understand patients as persons.[30] The concept of person is of vital importance in PCC as it transcends the more or less temporary and contextual role of being a patient. Persons have many features, characteristics and abilities in common, such as the ability to communicate and self-reflect, and at the same time, each person is someone in his or her own right with particular personal and circumstantial bodies, characteristics, abilities and histories.[31 32] For healthcare professionals (HCPs) to know their patients as persons, they need to be receptive to the unique premises of their patients' life. This approachability includes being sensitive to what is important to the patients, and neither diminish them on account of their vulnerability nor abandon them on account of their capabilities.[30] To support HCPs and organisations to enhance person-centredness in clinical practice the Gothenburg Centre for Person-Centred Care (GPCC) proposed a framework to operationalise the ethics of PCC.[29] In the framework a key factor in conducting PCC is the quality of communication and relationship, referred to as a partnership that includes evidence-based diagnostic procedures. Such a partnership is characterised by mutuality, respect, sharing and understanding, and is, at the same time, achieved through these means. Hence, a partnership is both the goal and a means to achieve PCC.

Previous evaluations of the GPCC framework in various conditions and care contexts have shown positive effects on several outcomes, including health-related quality of life (HRQoL),[33] patients' care experiences,[34] uncertainty in illness,[35] care costs[36 37] and self-efficacy.[26 27] However, this approach has not yet been evaluated for CMDs. Combining PCC with an eHealth tool was shown to enhance further the ability to improve self-efficacy,[38] and positive effects on self-efficacy have been demonstrated using telephone support.[28] In addition, it appears that developing a patient-professional partnership is not dependent on face-to-face encounters.[28 39] Telehealth and eHealth solutions challenge the traditional structures in healthcare and hold possibilities of increasing transparency and accessibility. eHealth services may increase the possibility of patient involvement (eg, empowerment and shared-decision making[40]), and psychological or behavioural eHealth interventions may, for example, contribute to a greater reach of patients who otherwise might not be subject to such assistance.[41] This paper presents the design of a study evaluating a person-centred eHealth intervention for patients on sick leave due to CMDs. The primary aim is to evaluate the effects of a person-centred eHealth intervention (digital platform and telephone support) for patients on sick leave due to CMDs. Secondary aims are to evaluate the cost-effectiveness and the process of implementation to assess the fidelity of the eHealth intervention.

## METHODS AND ANALYSIS
### Study design
The study is an open randomised controlled trial (RCT) with 1:1 allocation to either a control (standard care) or an intervention group (standard care plus person-centred eHealth intervention). Process evaluation and health economic analysis will also be conducted. The protocol is based on the Standard Protocol Items: Recommendations for Interventional Trials (SPIRIT)[42] guidelines and the results will be reported according to established recommendations.[43–45]

### Study context
The study takes place in a larger city area in Western Sweden. Nine public primary healthcare centres are participating, all of which are located in a socioeconomically diverse area. The social insurance system in Sweden allows employees to take a maximum sick leave of 7 days without a medical certificate, and except for an initial day of 'quarantine', they are financially covered by their employer during the first 14 days of illness. From day 8, a medical certificate is required that is usually obtained from a primary care or occupational healthcare physician. After 14 days of sick leave, benefits can be granted from the governmental Social Insurance Agency, which bases its decision on the medical certificate.

### Participants and recruitment
Recruitment of participants started in the spring of 2018 and is estimated to be completed during the fall of 2020. Patients on sick leave are consecutively screened in medical records by designated HCPs. Patients are eligible if they are on sick leave for no longer than 30 days due to a CMD that has been diagnosed by a physician. Table 1 presents the full inclusion and exclusion criteria. In the first step, eligible patients receive a letter with brief information about the study. Next, they are invited to make

**Table 1** Inclusion and exclusion criteria

| Inclusion criteria | Exclusion criteria |
|---|---|
| ► Patients aged 18–65 years<br>► Currently employed or studying at least part-time during the past 9 months<br>► Understand written and spoken Swedish<br>► Has a registered address<br>► Currently on sick leave that has not exceeded 30 days, primarily due to any of the following diagnoses in ICD-10 as diagnosed by a physician: mild-to-moderate depression (F32 and F33), mild-to-moderate anxiety disorder (F41), reaction to severe stress and adjustment disorders (F43, except post-traumatic stress disorder. F43.8A exhaustion disorder, a diagnosis in the Swedish ICD-10, is included) | ► Sick leave >14 days due to depression, anxiety disorders or stress reactions and disorders during the past 3 months<br>► Severe impairments hindering use of telephone or the digital platform<br>► Ongoing alcohol or drug abuse<br>► Any severe disease with an expected survival of <12 months or that can interfere with follow-up, or if the intervention is assessed as a burden<br>► Participating in a conflicting study |

ICD-10, International Classification of Diseases, Tenth Revision.

contact for more information about the study or they will be called within the upcoming week for further information. Patients willing to participate will be sent more detailed information about the study and a consent form by mail. After written consent has been received, the patient is randomised to either the control or intervention group and informed of the allocation. The randomisation is based on computer-generated random numbers and stratified by age (<50 or ≥50 years) and diagnostic group (depression, anxiety disorders or stress reactions and disorders).

### Patient and public involvement

Public involvement is an essential part of the research conducted at GPCC. The development of the digital platform was guided by a participatory design.[46] An advisory group comprising potential end-users (eg, patient-representatives and HCPs) offered suggestions to the design of the digital platform. These end-users will be continuously consulted throughout the study period (eg, in developing an interview guide and discussing dissemination).

### Control group

Standard care commonly involves meetings with a physician to follow-up on decisions regarding sick leave and RTW, as well as advice and information on self-care, but may encompass medication or cognitive behavioural therapy.[10] Referral to other healthcare professions depends on individual assessment and can include meetings with a physiotherapist, psychologist or psychotherapist, occupational therapist, group treatments, etc. Involving the workplace in an early, but timely stage is important regarding RTW.

### Intervention group

In addition to standard care, patients allocated to the intervention group will receive PCC via telephone and a digital platform. The intervention components are described in table 2. Following allocation, patients will be emailed a link to create an account and access the digital platform. They will be contacted by an HCP to schedule a time for an initial PCC conversation by telephone. This conversation, which is preferably scheduled within 1 week after initial contact, aspires to create a jointly agreed health plan, one that will serve as a foundation for future collaboration during the intervention period. A team of healthcare personnel with different professional backgrounds (eg, registered nurses and physiotherapists) are involved in conducting the intervention. All received a half-day introduction to CMDs and the philosophical underpinnings of PCC, and continuous training in how to apply PCC in practice (eg, through active listening, asking open-ended questions, jointly reflecting and providing summaries). They also have a forum to meet regularly with each other and specialists in the area throughout the intervention in which they can raise questions about the intervention as well as the practice of conducting care based on person-centred ethics. In this forum the HCPs will also share and discuss some of the health plans and telephone calls they have conducted, as part of their training and continuous development of skills in person-centred communication.

### Telephone support

In the telephone conversations the patient narrative is central and the HCPs will be attentive to the expressions of needs, goals, resources and experiences of the patient's current situation as well as their health status.[26 28] There is no fixed manual for how these conversations should develop nor any topics which they must cover. In each conversation, HCPs are asking questions and listening to the patient with the intent to understand the patient's experiences of their condition in the context of their everyday life. Together with the patient they discuss what could be achievable and desirable goals for the near future. They collaborate with the patient to identify strengths and resources, for example, by inquiring on how the patient has previously handled challenging situations. The patient's narrative is then documented in a health

**Table 2** Content of intervention components

| Telephone | Digital platform |
|---|---|
| ► HCPs initiate a first telephone conversation shortly after inclusion | On the platform, patients can:<br>► Write and read health plans<br>► Communicate with HCPs in a chat-like forum<br>► Seek information on their condition<br>► Invite significant others, such as family members and workplace representatives, to take part of their platform content<br>► Make daily notes and ratings on symptoms and general well-being, and visualise answers in the form of trend graphs |
| ► Following telephone communication is scheduled according to agreement but patients can also contact HCPs spontaneously | ► Patients can decide which functions of the platform they want the invited persons to access, and can both add new and delete persons at all times |
| ► The content of the telephone conversation(s) is central to what is documented in the patient's health plan, which is uploaded to the platform | ► If needed, HCPs will guide patients on how to access the platform. Patients are encouraged but not obliged to use the platform |

HCPs, healthcare professionals.

plan by either the patient or the HCP, and uploaded to the digital platform. The health plan should capture the patient's experiences of their situation and what health-related changes they want to achieve and how, including what resources they have in terms of personal capabilities and surrounding support systems. This health plan will serve as a guide in future contacts between the HCPs and the patient. It can be modified or reformulated according to what occurs during the intervention and the patient's process. Hence, the health plan is a living document negotiated in collaboration and documented for both the HCPs and patients (and preferably the extended network of the patient's choice) to have continuous access. This is what all patients participating in the intervention will receive, at a minimum, and this procedure builds on the GPCC framework to PCC.[29] After this first telephone call, following contact is discussed and planned in a mutual agreement between the HCP and the patient. Different HCPs can interact with the same patient during the intervention, depending on scheduling matters. If that is the case, the documentation of the health plan enables a continuity, so the patients do not have to restart their narrative process. However, if patients explicitly wish to continue interacting with the same HCP this is arranged for, if possible.

### The digital platform

The digital platform is constructed to be non-directive and create possibilities for patients to take an active part in their recovery and rehabilitation, which is an important objective in PCC. Initially, patients and HCPs have access to the platform. Patients can also choose to expand their network by inviting people of importance to them or their rehabilitation, such as family members, other healthcare contacts or workplace representatives. The platform should function as a mediator through which the patients, together with their network, can monitor symptoms and

be informed of their recovery process or risk of relapse or deterioration. The platform provides patients with the possibility to make daily ratings on a scale from 1-5 on symptoms and general well-being, for example how well they have slept or their ability to concentrate. The ratings will be visualised as graphs providing the possibility to see trends and development over time. The patients can take private notes about their ratings and hence this may function in a diary-like manner. In addition, the patients and the HCPs can communicate with each other through messages in a chat-like forum and the platform contains an assembly of links to other web pages on CMD that the patients can use to seek information or connections. HCPs log in to the platform at least once a day to be updated on patients' activities and check for messages. When patients invite someone to their platform page, they can decide which functions of the platform they want the invited person to access. Patients can both add new persons or delete persons that have access to the account. The platform can be accessed from any device with an internet connection and web browser, such as a computer, smartphone or iPad.

### Outcomes and measurements

Questionnaire data are gathered at baseline and after 3, 6, 12, 18 and 24 months from the inclusion date. Self-reported baseline characteristics of the participants' sex, age, civil status, country of birth, level of education, occupation and number of working years are collected through questionnaires sent by mail. Additional baseline characteristics (eg, income) will be gathered from the Longitudinal Integration Database for Health Insurance and Labour Market Studies.

### Primary outcome

The primary outcome measure is a composite score of changes in sick leave and general self-efficacy.[26 28 47] Patients

will be classified as improved, deteriorated or unchanged according to the following standards: if the patient has a reduced sick leave percentage and an increased general self-efficacy by ≥5 units at the 6-month follow-up, the patient is classified as improved. If the patient has an increased percentage of sick leave absence or a reduced general self-efficacy by ≥5 units at the 6-month follow-up, the patient is classified as deteriorated. Those who are neither deteriorated nor improved are considered unchanged. Sick leave will be regarded from two perspectives: the duration of the initial absence and the total absence after 24 months from the inclusion date. Data on full-time and part-time sick leave (25/50/75% of full-time) will be both self-reported by the participants through questionnaires and gathered from the Micro Data for the Analysis of Social Insurance register (MiDAS). Self-efficacy will be measured using the General Self-Efficacy Scale,[48 49] a 10-item self-assessment questionnaire measuring a person's general sense of competence towards dealing with unforeseen situations. Responses are made on a four-point scale (1=not at all true, 2=hardly true, 3=moderately true, 4=exactly true), with a composite score based on the sum of all items. The total score ranges from 10 to 40, with higher scores indicating a higher sense of self-efficacy.

### Secondary outcomes

Additional analyses will be conducted and based on data from the following instruments:
► General self-efficacy[48 49];
► EuroQol 5-Dimension health state questionnaire (EQ-5D)[50];
► Shirom-Melamed Burnout Questionnaire[51];
► Perceived Stress Scale[52];
► Self-rated Exhaustion Disorder[53];
► Hospital Anxiety and Depression Scale[54];
► Multidimensional Fatigue Symptom Inventory[55];
► Self-Efficacy for Managing Chronic Disease 6-item Scale[56];
► Sheehan Disability Scale[57];
► Healthcare utilisation;
► Incremental cost-utility ratios;
► Sick leave.

### Power/Sample size

To achieve a power of 80% based on an alpha error of 0.05 91 participants would be needed in each study group to detect an improvement in the composite score (20% in the control group vs 40% in the intervention group). We aim to include a minimum of 110 patients per group to have some margin for dropouts or withdrawals.

### Data analysis

Descriptive and analytic statistics will be conducted to compare the study groups. Logistic regression will be used on the primary outcome measure to calculate ORs with 95% CIs. For highly skewed data (eg, cost data), a bootstrap method will be used to estimate CIs.[58]

Parametric and non-parametric statistics will be carried out for secondary outcome measures. Intention-to-treat and, if needed, per-protocol analyses will be performed. Subgroup analyses will be conducted on, for example, sex, age and diagnostic groups.

### Economic evaluation

The economic evaluation will assess whether the cost of introducing this add-on PCC intervention is acceptable compared with the achieved health effects, for example measured as HRQoL collected through questionnaires. Missing questionnaire data will be assumed missing-at-random and thus handled using multiple imputations. The health state index derived from the EQ-5D questionnaire will be translated into quality-adjusted life-years using an area under the curve calculation and the Swedish experience-based valuation[59] and, as a sensitivity analysis, the society-based valuation for the UK.[60] The calculation of costs from a societal perspective will include costs for prescribed drugs, healthcare encounters and lost productivity. Drug costs will be obtained from the Swedish Prescribed Drug Register, held by the National Board of Health and Welfare. Healthcare encounters will be obtained from the regional patient register VEGA and converted to costs using diagnosis-related group (DRG) weights (available for all encounters in specialised healthcare) and the national cost per one DRG, whereas unit costs presented in national statistics will be used for primary care contacts (since DRGs are not available for primary care). In addition, we will collect information about the time spent by HCPs interacting with participants on the platform and by phone. Lost productivity will be obtained from both questionnaires and the MiDAS register, under the authority of the Social Insurance Agency, to ensure all absenteeism is recorded. This is done because the national register only fully covers days off from work that are reimbursed by the agency. Indirect costs for lost productivity will then be calculated using 'the human capital approach', which includes multiplying time off from work by mean wages (by age groups) and the social security contribution paid by employers.[61] Total costs and costs per payer category (patients, market sector, healthcare providers/regions and other authorities/tax-based funds) will be reported. The incremental cost will then be compared with incremental health effects using the incremental cost-effectiveness ratio (ICER). To explore the value of the intervention the ICER will be compared with the informal cost-effectiveness threshold used in Sweden of SEK 500 000 and to alternative willingness-to-pay thresholds using the cost-effectiveness acceptability curve.

### Process evaluation

To link the outcomes of the intervention to the implementation process we will explore three complementary aspects of the intervention: (1) how much and in what way participants have used the PCC intervention, (2) how they experienced it and (3) whether it has been used

as intended.[62] Quantitative data will be collected from survey questions and ratings of the telephone and platform support, which will be attached to the 3-month and 6-month questionnaires sent to those in the intervention group. We will also collect data on the use of different functions of the platform and number and modes of contact between HCPs, patients and persons invited to the platform during the intervention period. Qualitative data will consist of transcribed audio recordings of telephone conversations between HCPs and patients. In addition, open questions in the 3-month and 6-month surveys permit elaboration on experiences of the activities and delivery of the intervention. We will also interview a sample of intervention participants for an in-depth exploration of their experiences of the intervention. Qualitative data will be analysed using qualitative analysis methods and quantitative data using descriptive statistics. Using both quantitative and qualitative data allows applying mixed-methods analysis to triangulate the data, which increases the reliability of the study outcome.

## Ethics and dissemination

The study was approved by the Regional Ethical Review Board in Gothenburg, Sweden (DNr 497-17, T 023-18, T 526-18). All participants agreed to participate and signed a written consent form after receiving written and oral information about the study. Informed consent will also be asked of participants accepting to be interviewed. Studies in the project will be published in peer-reviewed scientific journals and presented at national and international conferences. This project is part of a research programme at the GPCC, where extensive work is ongoing to disseminate knowledge on and implementation of PCC.

## DISCUSSION

The person-centred healthcare process is based in a continuous negotiation to understand the general and unique premises of each patient's situation, with decisions and actions guided by that understanding.[30 63] In the present study, this is operationalised in the communication between HCPs and patients. The objective of the telephone conversations is for the patients to communicate their perspective regarding their condition and its effect on everyday life. The HCPs engage in the patient's narrative by active listening and asking open-ended questions, and together they set goals derived from this communication.[29] Consequently, PCC and person-centred interventions are complex, as they are individually tailored rather than following a standardised procedure. In the present study, the patients will use the different features of the intervention according to their individual needs. How the intervention has been conducted will be monitored and clarified in the process evaluation, which is an essential exploration in order to understand the pathways of a complex intervention.[62] Hawe et al[64] suggest that process evaluations of complex interventions needs to clearly state what is assumed to be the essential element of change. As

long as the essential elements are not altered, other intervention components can be adapted to serve the local context. Applied to the present study, we assume that the essential element to generate change is the quality of the person-centred communication. Considering the population of primary care patients on sick leave due to a CMD, part of the recent increase in mental illness-related sick leave is possibly explained by both work-related and private life stress exposure.[65] Psychosocial stress exposure can be conceptualised as an imbalance between demands, support and resources such as control, including both work and private life situation.[66] Consequently, we argue that it is essential with a person-centred communication rooted in an understanding of the patient as both vulnerable and capable and that reaching agreements on care and rehabilitation build on the knowledge of the patient's circumstances, needs, wishes and resources. We included self-efficacy as a primary outcome measure for the RCT. This choice is based mainly on two reasons. First, several studies have suggested self-efficacy as an important psychological resource in an RTW process.[20–22] Second, studies on PCC interventions have reported positive effects of self-efficacy.[26–28 38]

The study design has some limitations. Although the study is conducted in cooperation with primary care centres, the intervention is mainly performed by designated HCPs situated in a research centre in a separate location. One advantage of that arrangement is the possibility to follow the process closely thanks to full access to all patient-staff communication. However, the study results will have to be considered in light of that and future adaptions may be needed before implementing the intervention in other contexts and settings. Another limitation is the language restriction. It has not been feasible to involve interpreters and therefore only patients who can independently manage to communicate in Swedish will be included in the study. Participation also require access to an internet device. Furthermore, although we consider it important from a perspective of PCC not to impose a certain standardised procedure of the intervention on participants, it is also possible that participants will not use the intervention to its full extent, which we consider a limitation. For example, although earlier research show the importance of workplace involvement to impact on RTW,[15–19] this is included in the PROMISE intervention as a voluntary feature. Although the patients should be encouraged to invite their network to the platform, it is possible that the voluntary design will result in them doing so to a lesser degree than desired. Hopefully, the intervention components perceived as helpful by the participants, will also be applied to a desired extent.[67] It is also possible that the designation of a 5-point difference in general self-efficacy, which has proved relevant in other clinical settings,[26 28] will not be valid for common mental disorders.

The present study corresponds to the current call for interventions targeting sick leave in patients with CMDs. The design of the intervention is consistent with what

previous reviews have shown promising, for example, facilitating collaboration, using multiple components[19] and providing possibilities of involving the workplace.[15–19] The choice to develop and evaluate an eHealth intervention was based on the ambition to limit the number of physical appointments the patients would need to attend and to reach the patients in their everyday life at home, which represents some of the potential benefits of eHealth services.[40] However, it is fundamental that also remotely delivered healthcare is person-centred. Furthermore, we argue that developing interventions, which are not limited to a certain professional category and targeting not only one diagnosis, is necessary considering the current pressure on first-line healthcare facilities for CMDs. We consider it a strength that our study investigates PCC as an addition to, not in comparison with, standard care. This intervention approach is expected to gain new knowledge on how PCC can contribute to the challenges facing healthcare systems with a large number of persons seeking help for and on sick leave due to CMDs.

**Author affiliations**
¹Institute of Health and Care Sciences, Sahlgrenska Academy, University of Gothenburg, Gothenburg, Sweden
²Centre for Person-Centred Care (GPCC), University of Gothenburg, Gothenburg, Sweden
³Psychiatric department, Sahlgrenska University Hospital, Gothenburg, Sweden
⁴Department of Internal Medicine and Geriatrics, Sahlgrenska University Hospital Östra, Gothenburg, Sweden
⁵The Institute of Stress Medicine, Region Västra Götaland, Gothenburg, Sweden
⁶School of Public Health and Community Medicine, Institute of Medicine, University of Gothenburg, Gothenburg, Sweden
⁷Department of Molecular and Clinical Medicine, Sahlgrenska Academy, University of Gothenburg, Gothenburg, Sweden
⁸Research and Development, Primary Health Care, Region Västra Götaland, Gothenburg, Sweden

**Contributors** All authors were involved in the design of the study. MC drafted the manuscript with critical input from LA, IE, KG, IHJ, HG, KS and AF. AF is the principal investigator and grant holder of the investigation. All authors reviewed, edited and approved the final version of the manuscript.

**Funding** This work was supported by The Swedish Research Council for Health, Working Life and Welfare (reference number 2016-07418, 2017-00557 and 2019-01726). The study was financed by grants from the Swedish state under the agreement between the Swedish government and the country councils, the ALF agreement (ALFGBG-772191 and ALFGBG-932659).

**Competing interests** None declared.

**Patient and public involvement** Patients and/or the public were involved in the design, or conduct, or reporting, or dissemination plans of this research. Refer to the 'Methods' section for further details.

**Patient consent for publication** Not required.

**Provenance and peer review** Not commissioned; externally peer reviewed.

**ORCID iDs**
Matilda Cederberg http://orcid.org/0000-0003-4727-9638
Lilas Ali http://orcid.org/0000-0001-7027-4371
Hanna Gyllensten http://orcid.org/0000-0001-6890-5162
Andreas Fors http://orcid.org/0000-0001-8980-0538

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
