## [Reviewer comments · BMJ Open]

ARTICLE DETAILS

TITLE (PROVISIONAL)	A person-centred eHealth intervention for patients on sick leave due to common mental disorders: Protocol of a randomised controlled trial and process evaluation (PROMISE)
AUTHORS	Cederberg, Matilda; Ali, Lilas; Ekman, Inger; Glise, Kristina; Jonsdottir, Ingibjörg; Gyllensten, Hanna; Swedberg, Karl; Fors, Andreas

VERSION 1 – REVIEW

REVIEWER	Irene Jaén Parrilla Universitat Jaume I, Spain
REVIEW RETURNED	20-Feb-2020

GENERAL COMMENTS	This paper describes a protocol of a study aimed to test the efficacy of a digital platform (eHealth) plus telephone calls. The protocol is really interesting, well-written, and well-designed. However, I have a few suggestions for the paper that can add to the clarity of presentation. I will list my suggestions and questions below. I will hopefully enhance the quality of this study to increase the likelihood of acceptance: 1-How much time do professionals spend on the digital platform or how often do health professionals check patient activity on this platform? Are they instructed to check the patient's activity? 2- You mentioned that "team of health care personnel with different professional backgrounds (e.g., registered nurses and physiotherapists) are involved in conducting the intervention. All received a half-day course in CMD and training in the philosophical underpinnings of PCC and how to apply these in practice". Do you think there can be differences depending on the professional who makes the telephone call or give feedback through the digital platform? In relation to this, is the same professional who respond all times to the same patient or are professionals changing depending on their availability? 3- In the design, it was unclear to me how is the intervention that they receive by both phone and digital platform. How the digital platform is contributing to the implementation PCC? It may be helpful if you add more information about PCC in the main text and/or pictures of the digital platform. 4- If "there is no fixed manual for how these conversations should develop nor any topics which they must cover", how are you sure that the intervention is being carried out from a PCC approach? It would be advisable if you could explain a little more what the intervention consists of, and more specifically, what content or skills
---

	are trained in these half-day course. In my opinion, it would help readers to gain a better understanding of the intervention. 5- In the discussion section, you mentioned that you “intend to further explore the connection between PCC and self-efficacy, how are you going to do it? 6-My major concern in about adherence to the digital platform. You mentioned that the digital platform creates possibilities for patients to take an active part in their recovery and rehabilitation. What measures have you taken to patients take this active part, are you improving the adherence to the platform in some way? Are you telling patients when they have to log in to the digital platform (daily, weekly) or do they take the initiative?
--	---

REVIEWER	Frederieke Schaafsma Amsterdam UMC, The Netherlands
REVIEW RETURNED	01-Mar-2020

GENERAL COMMENTS	This protocol study aims to assess the process and effectiveness of a person-centred eHealth interventions for patients on sick leave due to CMD's. Person centred care refers to partnership between a health care professional and a patient and emphasizes the relevance of communication and relationship. This is a rather innovative view on health care, particularly within the field of occupational health care and guidance for return to work of workers on sick leave. I believe it is important that this trial is carried out. I have some comments or suggestions.  1. Are there any references available that show that a half day of training would suffice to achieve the goal of PCC (partnership) by the involved HCPs in this study? 2. Will there be an assessment of the quality of the HCPs offering the person centred care? 3. In the process evaluation the authors focus on the perceived effect and quality of the guidance from the perspective of the patients or HCP themselves, but what about the objective quality of the provided care? Are there any criteria for that? And if not, is this considered a limitation? 4. If I understand correctly the digital platform can also be made available to others e.g. family or partner. But the role of workplace remains somewhat unclear in this respect. Although the authors in the Introduction section and Discussion section report that the support from the workplace is essential, their role in this intervention is somewhat vague or not existent? If that is the case, this should be discussed in the Discussion section. 5. The digital platform is that a website accessible via laptop or PC. Or could this also be assessed on a mobile devise? I can imagine that for the majority of people (particularly younger ones), an app could improve compliance (lower the threshold to score how you feel on a daily basis). 6. The composite score of self-efficacy and sick leave: the authors consider an improvement or deterioration when the
---

	score on the GSE has changed for 5 points. On what basis is this change score considered a clinically relevant change? I have a similar question related to the power calculation. On what basis do you expect a 20% difference between intervention and control group? Is there any scientific literature available to substantiate this expectation?
--	--

VERSION 1 – AUTHOR RESPONSE

Reviewer: 1 Irene Jaén Parrilla
 Institution and Country: Universitat Jaume I, Spain

This paper describes a protocol of a study aimed to test the efficacy of a digital platform (eHealth) plus telephone calls. The protocol is really interesting, well-written, and well-designed. However, I have a few suggestions for the paper that can add to the clarity of presentation. I will list my suggestions and questions below. I will hopefully enhance the quality of this study to increase the likelihood of acceptance:

1-How much time do professionals spend on the digital platform or how often do health professionals check patient activity on this platform? Are they instructed to check the patient's activity?

The HCPs log in to the platform at least once but often several times per day, in order to check messages. At least once a day, or in relation to scheduled telephone calls, they check patients' activities on the platform. They also use the platform to write and upload health plans, which is done in connection to telephone conversations. When patients are new to the platform, during the first few weeks, if they do not spontaneously engage in the platform activities or if they engage initially but don't continue, HCPs can write a message to the patient in order to encourage them to be active. How much time they spend on the platform is to some extent dependent on how much the patients use the platform. This is an important aspect when addressing the cost-effectiveness of the intervention, as well as the process evaluation, and information about time spent will be included in those analyses.

At page 10-11, under the headline "The digital platform", a sentence describing a minimum HCP engagement with the platform has been added for clarification.

2- You mentioned that "team of health care personnel with different professional backgrounds (e.g., registered nurses and physiotherapists) are involved in conducting the intervention. All received a half-day course in CMD and training in the philosophical underpinnings of PCC and how to apply these in practice". Do you think there can be differences depending on the professional who makes the telephone call or give feedback through the digital platform? In relation to this, is the same professional who respond all times to the same patient or are professionals changing depending on their availability?

Thank you for raising this important issue. We expect that to some extent there will be differences in the communication depending on which professional conducts the telephone call. Eliciting the patient's narrative and engaging in a conversation concerning their situation is a communicative process requiring sensitivity and skills, but like each patient is a person with their unique characteristic, so is each HCP, and this will colour the conversations. However, all HCPs will engage in the conversations with the same agenda, according to the GPCC framework, of narrative elicitation,

creation of a patient-professional partnership and documentation of a health plan. In order to enhance clarification on this important issue, we have elaborated the description of the telephone conversations on page 10, and added a section at page 16-17 where we discuss this question.

Different HCPs can interact with the same patient during the intervention, depending on scheduling matters. We have added this information to the protocol at page 10.

3- In the design, it was unclear to me how is the intervention that they receive by both phone and digital platform. How the digital platform is contributing to the implementation PCC? It may be helpful if you add more information about PCC in the main text and/or pictures of the digital platform.

The digital platform enables patients to be more active in monitoring their day-to-day symptoms and well-being. It also enables for the patient to have all information regarding their condition, hence increasing transparency and sharing of responsibilities between HCPs and patients. These are important objectives of PCC, and is part of what constitutes a patient-professional partnership, which is explained in the background section in relation to the GPCC framework (page 4). The link between the objectives of the platform and PCC has been clarified in the opening sentence in “the digital platform” section (page 11). In addition, this has also been clarified in the protocol by adding a table (page 9) of the intervention describing the intention with the telephone communication and the intention with the digital platform.

4- If “there is no fixed manual for how these conversations should develop nor any topics which they must cover”, how are you sure that the intervention is being carried out from a PCC approach? It would be advisable if you could explain a little more what the intervention consists of, and more specifically, what content or skills are trained in these half-day course. In my opinion, it would help readers to gain a better understanding of the intervention.

Standardizing the intervention to the level of having a fixed manual would be a major deviance from the ethics underpinning PCC. Not manualizing these conversations but letting them take form according to the persons taking part in them, is one way of allowing the intervention to be carried out from a PCC approach. For example, in each conversation, HCPs are asking questions and listening to the patient with the intent to understand the patient’s everyday life and experiences of their condition, which is then documented in a health plan and uploaded to the platform. The health plan serves as an indicator of person-centred care and captures the communication and how this is enacted from a person-centred perspective. This is what all patients participating in the intervention will receive, at a minimum, and this procedure builds upon the GPCC framework to PCC, which has been described and is referred to in the protocol. In order to help the readers to gain a better understanding of the intervention, we have added a table (page 9) in order to present a clearer picture of what the intervention consist of. At page 10, we have developed the description of the telephone conversations and clarified what is a minimum level of PCC that the patients will receive. On page 8-9, we have added examples on the content of training in PCC.

5- In the discussion section, you mentioned that you “intend to further explore the connection between PCC and self-efficacy, how are you going to do it?”

This sentence was actually meant to refer to the planned subgroup analyses. We understand that this sentence creates confusion, and hence, we have deleted it from the discussion.

6- My major concern is about adherence to the digital platform. You mentioned that the digital platform creates possibilities for patients to take an active part in their recovery and rehabilitation. What measures have you taken to patients take this active part, are you improving the adherence to the platform in some way? Are you telling patients when they have to log in to the digital platform

(daily, weekly) or do they take the initiative

Measures to improve adherence have been taken into account particularly in the design of the intervention and platform, e.g., by involving representatives from patient communities to make sure that the platform corresponds to the needs of the stakeholders. One factor usually affecting adherence to internet-based or eHealth treatment in mental illness is HCP guidance. Our intervention is constructed so that many components involve communication with HCPs, and are interactive to their nature (such as the messages function, and the health plan). By messages on the platform as well as in the telephone conversations, HCPs can comment on patients' activities and encourage them to increase or continue their use of the platform, which has been added in Table 2 (page 9). However, how often a patient should use the platform is entirely up to the patient to decide. Rather than taking large measures to remind patients of the need to use the platform, we want to explore how useful they find the platform. We are monitoring the adherence to treatment components as a part of the process evaluation. Furthermore, it is probable that the intervention components which are perceived as helpful will be adhered to in a greater extent.

As we acknowledge that this topic raised by the reviewer is important, and that it is possible that our chosen demeanour will result in participants not using the platform in an optimal way, we have stated this as a limitation at page 18 and discuss it as such.

Reviewer 2: Frederieke Schaafsma

This protocol study aims to assess the process and effectiveness of a person-centred eHealth interventions for patients on sick leave due to CMD's. Person centred care refers to partnership between a health care professional and a patient and emphasizes the relevance of communication and relationship. This is a rather innovative view on health care, particularly within the field of occupational health care and guidance for return to work of workers on sick leave. I believe it is important that this trial is carried out

1. Are there any references available that show that a half day of training would suffice to achieve the goal of PCC (partnership) by the involved HCPs in this study?

We do not claim that a half-day training is sufficient to independently practice PCC, however in an earlier study on PCC a similar dose of HCP training succeeded in achieving the intended results (Fors et al, 2015). The half-day is to be considered an introduction to the philosophies underpinning PCC and the GPCC framework, which gives guidance on how to conduct person-centred care. As an introduction it covers relevant topics and discussions to understand what PCC is about, but the continuous reflective part (meetings in a group with more experienced professionals for example) of course deepens the understanding. Implementation of the GPCC framework should be done in accordance with the local context. In this study, the intervention is conducted in a research setting and a half-day introduction, followed by the regular meetings in the forum, was estimated to be a proportionate training given the results from earlier studies and the circumstances of the present study. We have elaborated the description on the HCP training on pages 8-9 in the protocol.

Fors, A., Ekman, I., Taft, C., Bjorkelund, C., Frid, K., Larsson, M. E., Thorn, J., Ulin, K., Wolf, A., Swedberg, K. (2015). Person-centred care after acute coronary syndrome, from hospital to primary care - A randomised controlled trial. *Int J Cardiol*, 187, 693-699. doi:10.1016/j.ijcard.2015.03.336

2. Will there be an assessment of the quality of the HCPs offering the person-centred care?

Since our intervention focus on person-centred care, what we will assess is the quality of the person-centeredness. This relates to the question of fidelity, which is a central concept in conducting process-evaluations. Here, assessment of quality or fidelity to PCC will not be performed in a quantitative manner but it will be part of the process evaluation, by exploring patient's perspectives on the quality of the person-centeredness through questionnaires and semi-structured interviews.

3. In the process evaluation the authors focus on the perceived effect and quality of the guidance from the perspective of the patients or HCP themselves, but what about the objective quality of the provided care? Are there any criteria for that? And if not, is this considered a limitation?

This is an important question. We agree that the quality of the provided care is an important aspect of the intervention. To enhance and safeguard the quality, the HCPs meet regularly with experts in the area. This is described in the protocol at pages 8-9 and we have expanded this description to make it more informative. In the process evaluation the aspect of fidelity will evaluate how the intervention has been delivered by analysing quantitative and qualitative data (e.g. audio-recordings of telephone conversations and data on how participants have perceived the communication and intervention components). Through this material, it will be possible for us to address aspects concerning quality of the intervention.

However, we do not in this project address the quality of care provided by the HCPs and do not evaluate the intervention in relation to indicators of quality according to evidence based practice. In future projects it would be interesting to evaluate whether there is a difference between caregivers conducting person-centred care and caregivers conducting care as usual in how well the care provided corresponds to best practice.

4. If I understand correctly the digital platform can also be made available to others e.g. family or partner. But the role of workplace remains somewhat unclear in this respect. Although the authors in the Introduction section and Discussion section report that the support from the workplace is essential, their role in this intervention is somewhat vague or not existent? If that is the case, this should be discussed in the Discussion section.

We agree with the reviewer that the role of the workplace have not been sufficiently clarified in the protocol, and we are thankful for this comment and opportunity to clarify. In designing the intervention, we have considered the previous research on interventions impacting sick leave and the importance of multiple components and workplace involvement. In our intervention, workplace involvement is facilitated by inviting workplace representatives to the platform and patients are informed of their possibility and the possible benefits. However, the final decision of who they want to involve in their health care process and platform is the patient's. This is mentioned in the protocol at page 11, in the description of the digital platform, but we have tried to further clarify it by adding a table of the characteristics of the intervention (table 2, page 9), in which it is stated that the patients can invite work place representatives to the platform. We have also reflected upon this as a possible limitation in the Discussion section at page 18.

5. The digital platform is that a website accessible via laptop or PC. Or could this also be assessed on a mobile devise? I can imagine that for the majority of people (particularly younger ones), an app could improve compliance (lower the threshold to score how you feel on a daily basis).

It is a website, but as such accessible through any device with an internet connection and a browser. We have added this description to the protocol at page 11. We advise the participants to create a bookmark of the website for easy access, this bookmark can be attached to the home screen. We

agree with the reviewer's comment that an app might improve compliance and the platform might be accessible as an app in the future.

6. The composite score of self-efficacy and sick leave: the authors consider an improvement or deterioration when the score on the GSE has changed for 5 points. On what basis is this change score considered a clinically relevant change? I have a similar question related to the power calculation. On what basis do you expect a 20% difference between intervention and control group? Is there any scientific literature available to substantiate this expectation?

Our designation of 5 points corresponds to the reported standard deviation, and derives from previous research also using a composite score as the primary outcome measure to evaluate effects of person-centred care (Fors et al, 2018; Fors et al, 2015). We discuss the choice of a 5 point-difference based on earlier studies conducted in other health care contexts as a possible limitation on page 18. The choice of a 20 % difference between groups is based on a pragmatic estimate what would be a clinically relevant difference for clinical practice (Fors et al, 2015).

Fors, A., Blanck, E., Ali, L., Ekberg-Jansson, A., Fu, M., Kjellberg Lindström, I., Mäkitalo, Å., Swedberg, K., Taft, C., Ekman, I. (2018). Effects of a person-centred telephone-support in patients with chronic obstructive pulmonary disease and/or chronic heart failure - A randomized controlled trial. Plos One, 13(8). doi:10.1371/journal.pone.0203031

Fors, A., Ekman, I., Taft, C., Bjorkelund, C., Frid, K., Larsson, M. E., Thorn, J., Ulin, K., Wolf, A., Swedberg, K. (2015). Person-centred care after acute coronary syndrome, from hospital to primary care - A randomised controlled trial. Int J Cardiol, 187, 693-699. doi:10.1016/j.ijcard.2015.03.336

VERSION 2 – REVIEW

REVIEWER	Irene Jaén Universitat Jaume I
REVIEW RETURNED	02-Jun-2020
GENERAL COMMENTS	The authors have adequately responded to the concerns raised. I congratulate them with a fine paper.
REVIEWER	F. Schaafsma Amsterdam UMC, The Netherlands
REVIEW RETURNED	19-Jun-2020
GENERAL COMMENTS	No comments.